# Multiple Traits Selection Strategies: A Proposal for Coffee Plant Breeding

Mateus Ribeiro Piza [1], Silvana Ramlow Otto Teixeira da Luz [2], Vinicius Teixeira Andrade [3], Vanessa Castro Figueiredo [3], Juliana Costa de Rezende Abrahão [3], Adriano Teodoro Bruzi [2] and Cesar Elias Botelho [3,*]

1 Department of Biology, Federal University of Lavras (UFLA), Lavras 37200-900, MG, Brazil; mateus.pr365@gmail.com
2 Department of Agriculture, Federal University of Lavras (UFLA), Lavras 37200-900, MG, Brazil; silvanaotto2016@gmail.com (S.R.O.T.d.L.); adrianobruzi@ufla.br (A.T.B.)
3 Southern Minas Regional Unit, Agricultural Research Corporation of Minas Gerais (Epamig), Lavras 37200-900, MG, Brazil; viniciustandrade.vta@gmail.com (V.T.A.); vcfigueiredo@epamig.br (V.C.F.); julianacosta@epamig.br (J.C.d.R.A.)
* Correspondence: cesarbotelho@epamig.br

**Abstract:** Experiments with progenies of perennial species such as coffee are generally affected by the heterogeneity of residual variances between information repeatedly collected in space and time on the same individual. In this study, we propose an index that considers the individual heritability of multiple traits for progeny selection and evaluate the applicability of this index in comparison with other indices in a real dataset. Data from 30 coffee genotypes in the $F_{4:5}$ generation were used to obtain the individual heritability values ($h_i^2$) of progenies that were subjected to factorial analysis to obtain the scores and construct a scatter plot, where graphical analysis (GA) was applied. Genetic gains were obtained for productivity and resistance to Cercospora leaf spot using GA. The best performance among the strategies ($-12.11\%$) was obtained using GA for resistance to Cercospora leaf spot, which has low heritability—contrary to the result obtained using the commonly used index based on the sum of Mulamba and Mock ranks. The GA approach allows an assertive selection to minimize the effects of heterogeneity between seasons, and greater genetic gains are obtained. Its use as a tool for the selection of perennial plant progenies based on multiple characters is promising.

**Keywords:** *Coffea arabica* L.; Cercospora leaf spots; individual heritability; genetic improvement; heterogeneous variance



## 1. Introduction

The search for more productive cultivars has become fundamental for those hoping to make production more sustainable and is a big challenge for genetic improvement programs. Close to 30 years is required to release a new coffee cultivar due to the need to evaluate the phenotypic expression of the genotypes over time, which is mainly completed during the juvenile stage of the coffee plant [1]. Thus, it is necessary to adopt effective and assertive methodologies to choose superior progenies for the agronomic traits of interest to maximize the potential of the field crop [2,3].

Experiments with progenies of perennial crops such as coffee generally require large areas and are generally affected by the heterogeneity of residual variances. This occurs by collecting information about the same genotype repeatedly in space and time, which can result in heterogeneous variances between measurements. In these cases, each genotype has accuracy, heritability, and residual variation values, requiring the adoption of analysis procedures that can accommodate heterogeneity without negatively affecting the parameters of interest [4,5].

The development of a superior and widely adapted cultivar requires the evaluation of multiple traits to be used as selection criteria. Selection indices have been used for the selection of superior progenies [6–8]. A numerical value is estimated as a function of the linear combination of multiple traits measured by the researcher. These traits have significant heritability and importance for simultaneous ranking and selection, in which weighting coefficients are estimated to maximize the correlation between the index and the true genetic value of the individual [9,10].

To calculate a selection index, genetic parameters, patterns such as heritability, and genetic or phenotypic correlations, can be used in association with the phenotypic variation of the data themselves, allowing for reliable estimates of the variance and covariance matrices and the prediction of nontarget populations not evaluated in experiments [11]. However, these indices have their particularities and limitations with respect to applications related to reporting the inaccuracy of estimates of variances and covariances [12] and estimates of traits of economic importance [13,14].

Multivariate analyses may show multicollinearity, leading to bias in the final result and difficulties in its interpretation [15]. In this sense, methods such as factor analysis and partial least squares can be adopted to work in a multiple trait scenario to incorporate variables with collinearity.

When using factor analysis, it is possible to form noncorrelated axes after factor rotation. Thus, multicollinearity is no longer an obstacle, allowing for the creation of unobserved latent variables as a function of the response variable. Most variance in the observed explanatory variables is concentrated in the first new latent variables, allowing for a reduction in dimensionality and simplification of the analysis [16].

The definition of the objective and the direction of selection in a breeding program are of fundamental importance; in addition, the appropriate criteria to be used by the breeder should suit these requirements and allow maximum accuracy and minimization of the associated biases. Thus, our aims were (i) to present an index that considers the individual heritability of multiple traits for the selection of genotypes and (ii) to evaluate the applicability of this index compared to other indices on a real dataset of a coffee progeny experiment.

## 2. Materials and Methods

Twenty-eight $F_{4:5}$ progenies and two commercial cultivars as controls (Catuaí Vermelho IAC 144 and MGS Aranãs) (Supplementary Materials Table S1) obtained by the breeding program at the Agricultural Research Corporation of Minas Gerais (Epamig) were used for the evaluation. These progenies are from the crosses between Icatu Amarelo IAC 2944 × Catuaí Amarelo IAC 62 and Icatu Amarelo IAC 2944 × IAC 5002, which were selected among the self-pollination generations. The genetic material IAC 5002 of red fruits refers to the backcrossing performed by the IAC between Catuaí Amarelo IAC H2077-2-12-70 and Mundo Novo IAC 515-20. The progenies were derived from the opening of the best progenies, evaluated based on multiple traits by the authors of [3].

The experiment was carried out at the Três Pontas Experimental Field (TPEF) located in the municipality of Três Pontas, Minas Gerais, Brazil, in a region of slightly undulating relief, with a slope of less than 20%, a latitude of 21°20′54.24″ south, and a longitude of 45°28′48.57″ west at 960 m above sea level.

During the field trails period, average temperatures of 20.18 and 20.50 °C with 1196.0 and 1351.0 mm of rainfall were observed for the agricultural years of 2019/2020 and 2020/2021, respectively (Três Pontas Experimental Field Station). Phytosanitary management was effective and timely, except for the chemical control of Cercospora leaf spots, aiming to identify and select the progenies resistant to *Cercospora coffeicola*.

A randomized complete block design with three replicates, totaling 90 plots, was used. The spacing was 3.6 m between rows and 0.7 m between plants, with eight plants per plot. The implementation and conduction of the experiment were informed by agronomic recommendations for the coffee crop.

### 2.1. Variables Analyzed

The following traits were evaluated in the 2019/2020 and 2020/2021 crop seasons prior to harvest: stem diameter (SD, cm, measured five centimeters from the ground, on the main stem), the number of plagiotropic branches in the main stem (NB), and vegetative vigor (VV), according to [17]. When the fruits reached the optimal level of ripeness (>80% of mature fruits), the harvest was conducted, and productivity (PROD, 60 kg bags per hectare) and yield (YI, liters per 60 kg bag) were evaluated. After drying (13% moisture content) and processing of the fruits, the percentage of high screen size (HS) was analyzed. The severity of Cercospora leaf spots was also evaluated. No occurrence of orange rust was observed in the experiment during the study period.

PROD was calculated based on the total fruit harvest, followed by the conversion to bags of 60 kg per hectare of processed coffee, according to the actual yield of each genotype. For the calculation of YI, samples of 4 L of coffee were used, collected by the total stripping of the fruits, placed in plastic nets and exposed to the sun until reaching approximately 11.0% water content, and then processed.

The VV was evaluated before harvesting at plant level per plot, with grades being assigned according to an arbitrary ten-point scale by three evaluators according to [17]. For the granulometric analyses, a sample of 300 g of processed raw grain was adopted and passed through a set of circular sieves and oblong sieves [18]. The weights of the grains retained in sieves 16, 17, 18, and 19 (high sieve, HS) were added and converted to percentages.

To determine the severity of Cercospora leaf spots, monthly evaluations were performed by three evaluators between January and June 2021, in which the third pair of leaves of two plagiotropic branches, selected randomly in the middle third of the cultivars plants studied, totaling eight leaves per plant and 48 per plot, were observed. To determine the severity, a diagrammatic scale with six levels was used [19], in which the area under the disease progress curve (AUDPC) was calculated for the severity of the brown eye spot (BES) [20].

### 2.2. Statistical Analyzes

2.2.1. Data Analysis

For the data analysis, mixed models were adopted to predict the genetic values, following the best linear unbiased prediction (BLUP) procedure with the estimation of the variance components obtained by the restricted maximum likelihood (REML) method. SELEGEN—REML/BLUP [21], as well as the appropriate models, were used for analysis. The variables were subjected to the likelihood ratio test (LRT) at the significance level of 0.05 probability, considering the chi-square approximation to the progeny mean level, to identify genetic variation between progenies and the potential to apply selection based on the parameters analyzed.

The joint and individual analyses of the data were performed after the analysis of normality of the residuals based on the Shapiro-Wilk test [22], and the detection of variance homogeneity between harvests was achieved in the maximum $F$ test [23] using the data collected at the plot level with the aid of the stats package in the R software version 4.1.3 [24].

The linear model, represented in equation $y = Xm + Zg + Wp + e$, was adopted for joint analysis of the data referring to SD, NB, VV, HS, YI, and PROD, where y is the data vector; m is the vector of the effects of the measurement–repetition combinations (assumed to be fixed) added to the mean; g is the vector of genotypic effects (assumed to be random, $g \sim N (0, \sigma^2)$); p is the vector of permanent environmental effects (plots) (assumed to be random, $p \sim N (0, \sigma^2)$); e is the vector of errors or residuals (assumed to be random, $e \sim N (0, \sigma^2)$); and X, Z, and W are the incidence matrices for these effects, respectively. The analysis of the AUDPC data for BES at the mean plot level was performed using the linear model represented in the equation $y = Xr + Zg + e$, where r is the vector of the repetition effects (assumed to be fixed) added to the overall mean.

For each characteristic, the following parameters were calculated: overall mean ($\mu$), genetic variance between progenies ($\sigma_g^2$), heritability of the mean of progenies in the broad sense ($h_a^2$), selective accuracy in the mean of progenies (*rap*), and the coefficient of genetic variation of progenies (CVg).

### 2.2.2. Selection Strategies

For each trait studied, the selection gain was estimated from direct selection (DS) and its percentage estimate (GDS, %) using R software version 4.1.3 and Genes [25]. Based on the sources of variation analyzed, the following selection indices were applied to perform progeny ranking:

i. The selection index of Mulamba and Mock [26], which consists of the sum of points (*Imm*) of the adjusted phenotypic means for each evaluated character. Two forms of weighting were assigned to this index. The first weighting (MM$_{(PE)}$) consisted of economic weights, with weight one assigned to the VV, HS, YI, and PROD variables and the weight of 0.8 assigned to the SD, NB, and BES variables. The second weighting (MM$_{(CVg)}$) was performed based on the coefficient of genetic variation of each trait.

ii. FAI-BLUP index [6], which is based on factor analysis and the genotype-ideotype distance.

iii. The index of the sum of standardized variables (ZI) [27]. The plot data were standardized, and then these values were subjected to a Scott-Knott test at 0.05 significance level.

### 2.2.3. Graphical Analysis Based on the Individual Heritability of Progeny

To promote selection based on multiple traits, we proposed the use of estimates of mean individual heritabilities ($h_i^2$) of progenies by the variance components (Individual REML) for the variables analyzed as selection parameters to accommodate the heterogeneities between progenies during the experiment and make the selection process more accurate [5].

The ($h_i^2$) values were estimated using Selegem–REML/BLUP software with the models described above and the equation $h_i^2 = \sigma_g^2 / \left[ \sigma_g^2 + (\sigma_{ei}^2/b) \right]$, where $h_i^2$ is the mean individual heritability of progeny $i$, $\sigma_g^2$ is the genetic variance between progenies, $\sigma_{ei}^2$ is the residual variance between plots for progeny $i$, and $b$ is the number of replicates.

From the values of $h_i^2$, an exploratory factor analysis was performed. Principal component analysis was used to extract the factor loadings from the correlation matrix between individual heritabilities. The varimax criterion [28] was used for the orthogonal analytical rotation. The scores of the genotypes were obtained using the ordinary least squares method, according to [29], with the equation $F = Z\left(A^T R^{-1}\right)^T$, where *F* is a $p \times f$ matrix with the factor scores representing the number of progenies ($p$) and the number of factors ($f$), Z is a $g \times s$ matrix with standardized individual heritabilities, A is an $s \times f$ matrix of canonical loadings, and R is an $s \times s$ correlation matrix between the variables, where $s$ represents the singular values.

The criterion to define the number of final factors was based on considering only those factors with eigenvalues greater than one [30]. The definition of the explanatory variables in each factor was based on factors that had a commonality value (proportion of the variation of the $i$-th variable that can be attributed to the $f$ common factors) and specific variance greater than 0.64.

Once the factor scores were obtained, they were plotted on a scatterplot using R to promote the selection of individuals who would maximize the genetic gains as a function of the ideotype of the individual heritability estimates from graphical analysis (GA). In the dispersion plane of the scores, axes were plotted as a function of the mean of the scores belonging to each factor, thus creating four quadrants.

The results were interpreted as follows: in quadrant I, the greatest gains as a function of individual heritability were obtained for the progenies, and it is possible to apply intensity of selection to the identified group, isolating the proportion of progenies with the highest scores. In quadrants II and IV, there were progenies that showed gains as a function of the

subregion determined by the main factor, with quadrants II and IV as the factors present in the y and x axes, respectively. In quadrant III, there are low-performance progenies which are not candidates for selection.

In situations where only two factors were adopted, the selection results were interpreted directly; however, with the need to deal with a greater number of explanatory factors, a sequence must be created with all combinations of factors, and a decision must be made based on a joint analysis of the graphs.

The data were analyzed with R software version 4.1.3 using the psych [31] and tidyverse [32] packages.

## 3. Results

### 3.1. Direct Selection

First, we estimated the genetic parameters (Table 1), and genetic variability was detected in all traits. The mean heritability of the genotype ranged from 41.8 to 74.9%. The accuracy in the mean of progenies varied between 0.85 (SD) and 0.67 (HS), showing the effectiveness of the inference regarding the estimated genotypic values in relation to the real genotypic values. The values observed for the coefficient of genetic variation of progenies were 7.77 to 32.01. These parameters represent the magnitude of genetic variability and indicate that the selection of superior progenies based on these traits will result in a significant increase in the genetic value of the population.

**Table 1.** Estimation of parameters for stem diameter (SD), number of plagiotropic branches (NB), vegetative vigor (VV), percentage of high sieve (HS), yield (YI), productivity (PROD), and area under the progress curve of the disease for the severity of Cercospora leaf spots (BES) for the selection of progenies of coffee plants in the $F_{4:5}$ generation.

| Parameter | SD | BN | VV | HS | YI | PROD | BES |
|---|---|---|---|---|---|---|---|
| $\mu$ | 11.78 | 45.98 | 6.63 | 47.57 | 3.02 | 3.04 | 17.02 |
| $\sigma_g^2$ | 16.44 | 22.19 | 0.23 | 26.69 | 0.10 | 0.18 | 18.43 |
| $h_a^2$ | 74.90 | 59.40 | 53.10 | 41.80 | 61.70 | 52.80 | 47.40 |
| $r_{ap}$ | 0.85 | 0.76 | 0.72 | 0.64 | 0.77 | 0.72 | 0.68 |
| CVg | 11.24 | 10.02 | 7.77 | 12.20 | 19.78 | 32.01 | 25.24 |
| LRT | 19.99 * | 8.44 * | 5.90 * | 3.96 * | 9.53 * | 5.81 * | 4.33 * |

Overall mean ($\mu$), genetic variance between progenies ($\sigma_g^2$), heritability of the mean of progenies in the broad sense ($h_a^2$), selective accuracy in the mean of progenies ($r_{ap}$), coefficient of genetic variation of progenies (CVg), and likelihood ratio test (LRT). * Significant at the 0.05 probability level, considering the chi-square approximation.

The SD with an intensity of 20% highlights the gains for the variables YI, PROD, and BES, with −18.28, 29.22, and −17.78% over the mean of the progenies and −20.80, 12.75, and −26.87% over the mean number of controls, respectively (Table 2), based on the ideotype. It is noteworthy that these variables have the highest CVg estimates (Table 1), confirming the selection accuracy. For HS, a gain in the opposite direction to the ideotype of −0.61% is observed when compared to the mean of the controls; however, the gain is positive in relation to the mean of the progenies (9.12%).

Based on the group of progenies individually selected for each trait, it was verified that progenies P14 and P16 maximized the gain for four variables, with the highest number of simultaneous traits. Progenies P2, P3, P19, P20, and P21 were selected for three traits; progenies P7, P18, P23, P24, and P26 were selected for two sources of variation; and the progenies P1, P4, P11, P13, P17, P25, and P27, along with the controls P29 and P30, were selected for one variable (Table 2). In general, 70% of the evaluated materials were selected for at least one trait studied, showing the high genetic potential of the population.

**Table 2.** Genotypes selected for each trait and gains from direct selection of individuals with an intensity above 20%. Ranking (R) of progenies, stem diameter (SD, millimeters), number of plagiotropic branches (NB), vegetative vigor (VV), sieve percentage of 16 and above (HS), yield (YI), productivity (PROD), and area under the disease progress curve for the severity of Cercospora leaf spots (BES).

| R | SD | BN | VV | HS | YI | PROD | BES |
|---|------|------|------|------|------|------|------|
| 1 | P19 | P14 | P19 | P3 | P7 | P26 | P23 |
| 2 | P24 | P1 | P3 | P29 | P23 | P19 | P16 |
| 3 | P26 | P18 | P14 | P2 | P2 | P24 | P18 |
| 4 | P27 | P16 | P21 | P4 | P21 | P20 | P14 |
| 5 | P20 | P13 | P17 | P30 | P25 | P21 | P7 |
| 6 | P14 | P20 | P16 | P16 | P11 | P3 | P2 |
| Mean Prog. | 39.726 | 64.269 | 6.617 | 46.905 | 587.142 | 33.164 | 16.527 |
| Mean Check | 37.042 | 66.427 | 7.096 | 57.995 | 623.313 | 41.496 | 23.842 |
| Mean SD | 45.721 | 72.786 | 7.350 | 57.144 | 413.142 | 51.516 | 10.326 |
| $h_a^2$ | 0.749 | 0.594 | 0.531 | 0.418 | 0.617 | 0.528 | 0.474 |
| GDS$_{(p)}$—% | 11.30 | 7.87 | 5.89 | 9.12 | −18.28 | 29.22 | −17.78 |
| GDS$_{(t)}$—% | 17.55 | 5.69 | 1.90 | −0.61 | −20.80 | 12.75 | −26.87 |

Progenies Mean (Mean Prog.), Check mean (Mean Check), Mean of the progenies selected per variable (Mean SD), mean heritability of progenies in the broad sense ($h_a^2$), percentage of selection gain relative to the mean of progenies (GDS$_{(p)}$, %), and percentage of selection gain relative to the mean of progenies (GDS$_{(t)}$, %).

### 3.2. Selection Indices

To simultaneously accumulate gains in multiple traits, the selection indices, including the index based on the sum of Mulamba and Mock ranks with prefixed economic weights (MM$_{(PE)}$), the index based on the sum of Mulamba and Mock ranks with economic weights based on the coefficient of genetic variation (MM$_{(CVg)}$), the FAI-BLUP index (FAI-BLUP), and the sum of standardized variables index (ZI) (Table 3). With a 20% selection proportion, there was coincidence between five progenies (P1, P3 P19, P20, and P21) that were selected based on using the two sum of positions strategies. There were direct gains for four traits, namely, NB, VV, HS, and BES, based on the selection of P16 using MM$_{(PE)}$. P24 is notable and was selected for two characteristics (SD and PROD) using MM$_{(CVg)}$. In general, the progenies that coincided with three characteristics for direct selection were selected based on the sum of two position strategies (MM$_{(PE)}$ and MM$_{(CVg)}$), with gains mainly for the variables SD, NB, VV, HS, and PROD.

**Table 3.** Ranking (R) of progenies (P) of coffee plants in the F$_{4:5}$ generation in terms of the mean of two harvests, the number of corresponding traits (C), and individual value of the index (VI) with the direct selection of progenies under different selection strategies with a 20% selection proportion.

| R | MM$_{(PE)}$ | | | MM$_{(CVg)}$ | | | FAI-BLUP | | | IZ | | |
|---|------|---|------|------|---|---------|------|---|------|------|---|---------|
| | P | C | IV | P | C | IV | P | C | IV | P | C | IV |
| 1 | P3 | 3 | 41.60 | P3 | 3 | 860.96 | P19 | 3 | 0.36 | P3 | 3 | 5.98 a* |
| 2 | P1 | 1 | 61.40 | P1 | 1 | 1098.93 | P24 | 2 | 0.35 | P19 | 3 | 5.90 a |
| 3 | P20 | 3 | 65.00 | P24 | 2 | 1100.05 | P21 | 3 | 0.27 | P24 | 2 | 5.42 a |
| 4 | P21 | 3 | 65.40 | P20 | 3 | 1123.65 | P20 | 3 | 0.26 | P21 | 3 | 5.10 a |
| 5 | P16 | 4 | 65.60 | P21 | 3 | 1186.61 | P26 | 2 | 0.24 | P1 | 1 | 4.58 a |
| 6 | P19 | 3 | 66.60 | P19 | 3 | 1282.99 | P27 | 1 | 0.22 | P20 | 3 | 3.66 a |

The index based on the sum of Mulamba and Mock ranks with prefixed economic weights (MM$_{(PE)}$), the index based on the sum of Mulamba and Mock ranks with economic weights based on the coefficient of genetic variation (MM$_{(CVg)}$), the FAI- BLUP (FAI-BLUP) index and the sum of standardized variables index (ZI). * Means followed by the same letter did not differ by the Scott-Knott test at the 0.05 significance level.

When using the FAI-BLUP index, progenies P19, P20, P21, P24, P26, and P27 were selected. SD, VV, and PROD were the variables that most coincided with the direct selection

in these progenies, directing the ranking. When using the ZI, the exact same progenies as those selected using $MM_{(CVg)}$ (P1, P3, P19, P20, P21, and P24) were selected. Only the ranking order was changed according to the methodologies inherent to each method. Progenies P3, P19, P20, and P21 were directly selected for three traits, coinciding with the variable PROD.

*3.3. Graphical Analysis*

The individual heritability values of progenies were obtained using the variance components by genotype (individual REML) for each trait studied. These values were analyzed to promote the ranking of genotypes in a multivariate manner.

Factor analysis was adopted for data analysis to reduce dimensionality as a function of the correlation between the adopted variables to explain the covariance between them [30]. Eigenvalue estimates greater than one were obtained for two factors, and this was the criterion adopted to determine the number of factors to obtain the final factor loadings (Figure 1). These two factors explain 59% of the total variation in the data. The value obtained by the Kaiser–Meyer–Olkin (KMO) statistic was 0.66, and Bartlett's test of sphericity was statistically significant (*p* value < 0.01), indicating that the factor analysis is adequate because the correlation matrix does not have a diagonal structure.

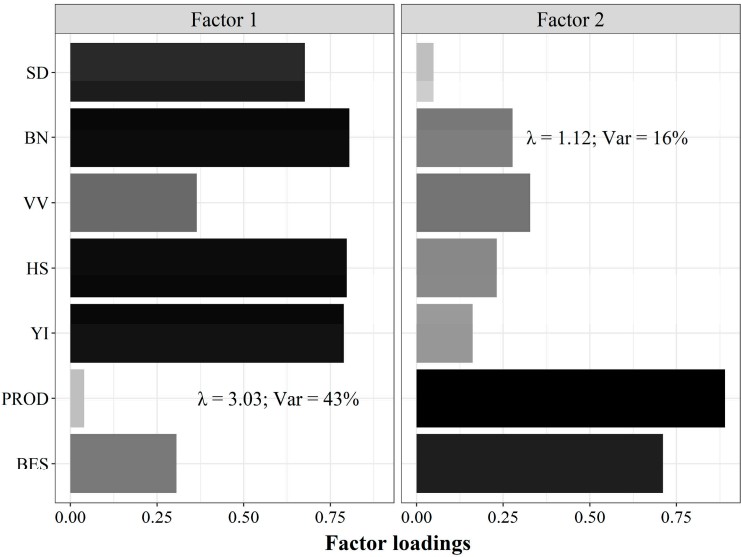

**Figure 1.** Final factor loadings of the factor analysis of individual heritabilities for the selection of superior progenies, estimate of eigenvalues (λ), and percentage of the total explained variation (Var). The greater the black color tone of the bars, the greater the correlation between the variables within each factor.

The dispersion of the scores generated by factors one and two is shown in Figure 2. The means of the scores obtained correspond to the vertical and horizontal transverse axes for factors one and two, respectively, resulting in the formation of four quadrants, promoting dissimilarity between the progenies to promote selection.

Quadrant one of Figure 2 shows the genotypes that jointly maximize the explanatory variables grouped into the two factors, thus favoring selection [33]. The progenies P1, P4, P8, P12, P14, P16, P18, P20, and P22, as well as the control P29, were selected. To standardize the 20% selection proportion, progenies P8, P12, P14, P16, P18, and P20, which have higher scores for both factors, were selected. In comparison to the selection indices, there was coincidence of selection for this group for progenies P16 and P20.

As shown in quadrant three, the progenies P5, P7, P9, P11, and P15 and the control P30 are grouped. These materials had the lowest gains from the individual analysis; thus, selection is not favored according to the proposal presented. In quadrants two and four, the progenies with characteristics inherent to the subregion generated by the main factor,

Factor 1 and Factor 2, respectively, are listed. Thus, the selection is directed to the most explanatory variables of each factor, which is not interesting in a multiple selection scenario (Figure 2).

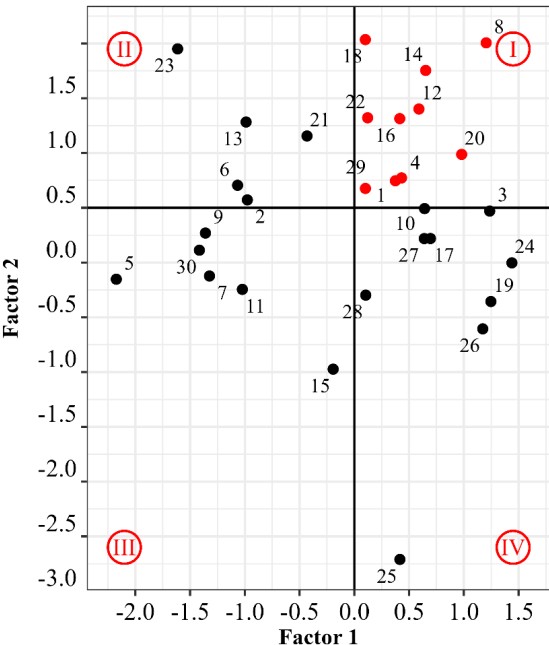

**Figure 2.** Graphical analysis of the individual heritability of twenty-eight progenies and two controls based on the scores obtained in the factor analysis, with a dimensionality reduction criterion based on n eigenvalues greater than one. I, II, III, and IV represent the genotype classification quadrants. The progenies that maximize gains for the created mega-factorial variables are shown in red. In black are the progenies that were not selected.

*3.4. Comparison between Selection Strategies*

From GA (Table 4), gains relative to the progenies selected were observed for YI, PROD, and BES, with the latter being of greater magnitude ($-12.11\%$) compared to the mean of the progenies. This was the strategy that resulted in the highest BES gain, which is associated with a positive gain in PROD due to the significant correlation at the individual heritability level of the progeny and of the same direction between the two variables (Supplementary Materials Table S2), in addition to high estimates of CVg.

**Table 4.** Gains from selection in relation to the mean of the progenies for the selection of superior individuals using graphical analysis and different selection indices for the stem diameter (SD, millimeters), number of plagiotropic branches (NB), vegetative vigor (VV), sieve percentage 16 and above (HS), yield (YI), productivity (PROD), and area under the progress curve of the disease to the severity of Cercospora leaf spots (BES).

| Parameters | | SD | NB | VV | HS | YI | PROD | BES |
|---|---|---|---|---|---|---|---|---|
| GA | Mean S. | 38.111 | 61.792 | 6.417 | 45.459 | 548.542 | 34.247 | 12.304 |
| | GS—% | −3.05 | −2.29 | −1.60 | −1.29 | −4.06 | 1.72 | −12.11 |
| MM(PE) | Mean S. | 38.110 | 64.065 | 6.615 | 52.897 | 570.148 | 38.694 | 14.259 |
| | GS—% | −3.05 | −0.19 | −0.02 | 5.34 | −1.79 | 8.81 | −6.50 |
| MM(CVg) | Mean S. | 37.777 | 62.353 | 6.504 | 51.981 | 538.552 | 38.052 | 14.724 |
| | GS—% | −3.68 | −1.77 | −0.91 | 4.52 | −5.11 | 7.78 | −5.17 |
| FAI-BLUP | Mean S. | 40.000 | 64.093 | 6.563 | 52.307 | 570.786 | 36.235 | 16.157 |
| | GS—% | 0.52 | −0.16 | −0.43 | 4.81 | −1.72 | 4.89 | −1.06 |

**Table 4.** *Cont.*

| | Parameters | SD | NB | VV | HS | YI | PROD | BES |
|---|---|---|---|---|---|---|---|---|
| ZI | Mean S. | 37.778 | 62.353 | 6.504 | 51.981 | 538.552 | 38.052 | 14.724 |
| | GS—% | −3.67 | −1.77 | −0.91 | 4.52 | −5.11 | 7.78 | −5.17 |
| | ha2 | 0.749 | 0.594 | 0.531 | 0.418 | 0.617 | 0.528 | 0.474 |
| | Mean P | 39.726 | 64.269 | 6.617 | 46.905 | 587.142 | 33.164 | 16.527 |

Mean of the progenies (Mean P), heritability of the mean of progenies in the broad sense ($h_a^2$), mean of the progenies selected by the different methods (Mean S), percentage of selection gain relative to the mean of progenies (GS, %). Graphical analysis (GA), the index based on the sum of Mulamba and Mock ranks with prefixed economic weights (MM$_{(PE)}$), the index based on the sum of Mulamba and Mock ranks with economic weights based on the coefficient of genetic variation (MM$_{(CVg)}$), the FAI-BLUP (FAI-BLUP) index, and the sum of standardized variables index (ZI).

The gains obtained using MM$_{(CVg)}$ and ZI were identical, and they selected the same genotypes. These gains were similar to those found using MM$_{(PE)}$, as observed for HS, YI, PROD, and BES. The use of the FAI-BLUP index maximized the selection of individuals with multiple traits of agronomic interest for the largest number of variables (SD, HS, YI, PROD, and BES).

To optimize the choice of the most efficient selection strategy, the percentage coincidence for the selected group was estimated on the fixed selection proportion, with direct selection (Figure 3a) based on practical interests, as well as the coincidence between the indices (Figure 3b). Based on SD, only the use of MM$_{(PE)}$ resulted in the selection of a group of progenies that had all the traits evaluated, facilitating the accumulation of favorable alleles in the population. The use of the FAI-BLUP index showed low selection efficiency for the BES and HS variables, resulting in gains of low magnitude for the ideotype.

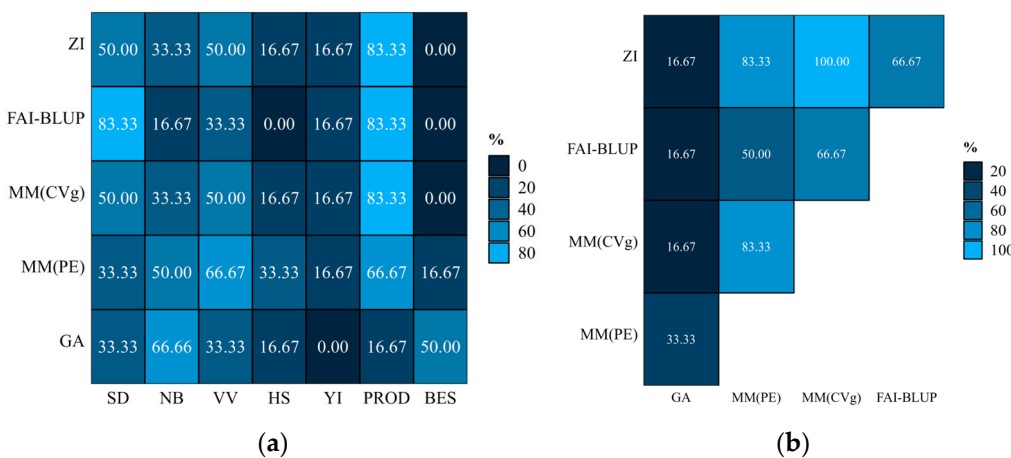

**Figure 3.** Percentage of coincidence between the direct selection for each adopted trait and the selection strategies (**a**) and coincidence between the selection indices (**b**) for the stem diameter (SD), number of plagiotropic branches (NB), vegetative vigor (VV), sieve percentage 16 and above (HS), yield (YI), productivity (PROD), and area under the disease progress curve for the severity of Cercospora leaf spots (BES). Index based on the sum of Mulamba and Mock ranks with prefixed economic weights (MM$_{(PE)}$), the index based on the sum of Mulamba and Mock ranks with economic weights based on the coefficient of genetic variation (MM$_{(CVg)}$), the FAI-BLUP (FAI-BLUP) index, and the sum of standardized variables index (ZI).

MM$_{(PE)}$ and GA were contrasting strategies in relation to the selection of the variables PROD and BES. The coincidence values for these variables when using MM$_{(PE)}$ and GA were 16.67% and 66.67% for PROD and 50.0% and 16.67% for BES, respectively (Figure 3a). This shows that the choice of a selection strategy implies targeting the improved population at the expense of specific characteristics. Among the indices, in Figure 3b, MM$_{(CVg)}$ was

100% coincident with ZI and 83.33% with MM$_{(EP)}$, reflecting equivalent strategies to be adopted for the selection of superior progenies in this dataset. The highest coincidence with MM$_{(PE)}$ was GA, while the lowest coincidence, with a value of 16.67%, was obtained between GA, the FAI-BLUP index, and ZI. These strategies promote the selection of groups of divergent progenies in this dataset.

According to the estimates of the Spearman correlations (Figure 4), with significance based on the t test at the 0.05 significance level, all selection strategies and the SD, estimated among the 20% selection proportion, there was an adjustment of one between ZI and MM$_{(CVg)}$ for all variables. On the other hand, GA was significantly correlated with MM$_{(PE)}$ for the variables SD (0.38) and in the opposite direction for VV (−0.43) and RD (−0.77).

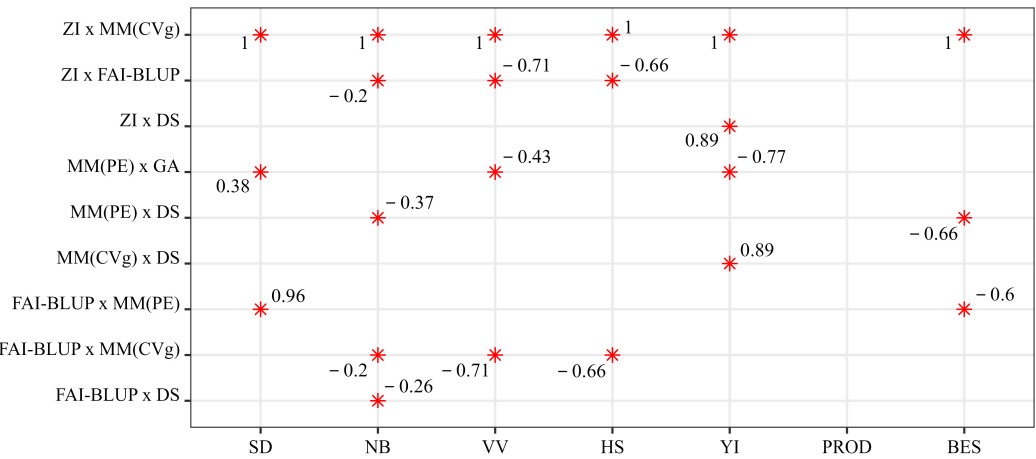

**Figure 4.** Spearman correlations, with significance based on the *t* test at the 0.05 significance level, between the different selection strategies for the sources of variation adopted. Index based on the sum of Mulamba and Mock ranks with prefixed economic weights (MM$_{(PE)}$), the index based on the sum of Mulamba and Mock ranks with economic weights based on the coefficient of genetic variation (MM$_{(CVg)}$), the FAI-BLUP (FAI-BLUP) index, and the sum of standardized variables index (ZI). Stem diameter (SD), number of plagiotropic branches (NB), vegetative vigor (VV), sieve percentage 16 and above (HS), yield (YI), productivity (PROD), and area under the disease progress curve for the severity of Cercospora leaf spots (BES).

Among ZI *vs.* FAI-BLUP, MM$_{(PE)}$ *vs.* FAI-BLUP, and MM$_{(CVg)}$ *vs.* FAI-BLUP, the correlations were significant and of moderate magnitude but in the opposite direction, diverging in the maximization of gains for SD, VV, and PROD. This fact was also evidenced for BES in relation to the correlations between MM$_{(PE)}$ *vs.* FAI-BLUP and MM$_{(PE)}$ *vs.* SD.

The lowest correlation estimates were obtained when analyzing the NB variable, where MM$_{(PE)}$ *vs.* SD, ZI *vs.* FAI-BLUP, SD *vs.* FAI-BLUP, and MM$_{(CVg)}$ *vs.* FAI-BLUP were significant but in the opposite direction, showing that the adoption of these strategies jointly leads to gains in opposite directions.

MM$_{(CVg)}$ *vs.* FAI-BLUP has a correlation of 0.96 for the selection of DS, thus proving its worth as a good tool for this variable, which has high heritability. For HS, the estimates were the same for ZI *vs.* FAI-BLUP and FAI-BLUP *vs.* MM$_{(CVg)}$ and in opposite directions (−0.66) with moderate magnitude for VG.

## 4. Discussion

For the success of a breeding program, it is essential that the selection strategies to obtain superior progenies are efficient for quantitative traits of low heritability in univariate or multivariate models so that generation advancements result in high genetic gains. The selection of multiple traits in perennial crops has been adopted because it allows for a more assertive identification of the genetic divergence of the progenies evaluated for the formation of a cultivar with various phenotypes of interest [34,35].

For the coffee crop, genetic selection is only efficient with the evaluation of several crops to verify the real performance of the studied progenies [36]. Among different harvests, genetic and environmental factors may result in the heterogeneity of genetic and environmental variances, reducing the accuracy of the prediction of the actual genetic value and, consequently, the response to selection; this may be more severe when the heterogeneity of the residual variance is accentuated [37,38].

In our study, we presented the GA methodology based on $h_i^2$, which allows selection as a function of the actual genetic value of the progenies over the years of evaluation in addition to exploring the correlation and covariance between the variables, accommodating the heterogeneity of the residual variances existing between harvests, in which the selection of superior progenies is performed by maximizing the scores obtained by factor analysis.

Factor analysis reasonably fit the dataset, as evidenced by the KMO value, which was greater than 0.5 [39], and by the significant Bartlett test. Ref. [40] reported that because factor analysis addresses phenotypic correlation matrices, it adequately modulates the relationship between variables for multivariate selection.

Regarding the final factor loadings, it can be seen that, in factor one, the traits NB, HS, and YI were grouped as explanatory, which indicates strong similarity between these characteristics (Figure 1). For factor 2, the PROD variable stood out, and a good relationship was observed with the BES variable. This grouping within and between factors occurs at the expense of the magnitude of the correlations observed between the variables—being considered strongly associated within the same factor and independent between factors [30].

In this database, only two factors had eigenvalues greater than one, and only one biplot was created to present the dispersion of the scores in relation to the variables analyzed (Figure 2). In situations where more factors are needed to provide a good explanation of the variance of the data, the creation of m biplots, in which all possible combinations of scores can be created to select progenies, should be created. For this situation, the method requires ranking the progenies with greater frequency in quadrant one to maximize the selection with multitrait gains based on the explanatory traits of each factor.

In this selection procedure, we identified PROD and BES, which have a negative genetic correlation (Supplementary Materials Table S3) of −0.64 and low heritability, positive gains of high magnitude, and good coincidence rates with direct selection. This shows the advantages of this methodology for low heritability traits, making selection more complex because they are controlled by several genes [41].

Among the indices for the selection of perennial plant progenies evaluated in this study, good performance for the selection of progenies was obtained using $MM_{(PE)}$, which has been used by several breeders as a selection strategy [3,42]. The great difficulty in using this and other methodologies is the definition of economic weights to maximize the correlation of genetic values, which lacks a standard procedure [14]. Some procedures, such as the use of genetic variation coefficients [42], heritability [43], or random weights [14], especially where weights are subject to estimation bias and impact the selection result, have already been tested. In the method proposed in the present study, there is no need to assign weights, which standardizes the selection procedure and makes it more accurate.

The results of this study demonstrate the efficiency of GA for the selection of coffee progenies, resulting in positive genetic gains based on the ideotype. This strategy is especially important for coffee—a perennial long-cycle species in which the development of new cultivars can take decades. Therefore, tools that can be utilized in the selection of progenies in a more assertive and accurate manner can accelerate the program and increase its efficiency, in addition to making it more competitive. Graphical analysis has its advantages and can be adopted as a selection tool for the culture of coffee plants and for other perennial species.

## 5. Conclusions

The use of individual heritabilities of progenies was efficient for the selection of superior progenies, and graphical analysis based on the factorial scores of these data allows for a more assertive selection, minimizing the effects of heterogeneity between harvests.

The adopted dispersion model better identifies the behavior of the genotypes, generating good interpretations of the behavior of the study population. Selection using characteristics of low heritability in the mean of progenies via GA based on the $h_i^2$ results in higher genetic gains compared to other multivariate strategies. In our study, it was possible to confirm that GA can be used as a tool for the selection of perennial plant progenies based on multiple agronomic traits.

**Supplementary Materials:** The following supporting information can be downloaded at: https://www.mdpi.com/article/10.3390/agronomy13082033/s1, Table S1—Relationship and characterization of the progenies used in the assay performed in Três Pontas, Minas Gerais; Table S2—Pearson correlation matrix between individual heritabilities for the selection of superior individuals; Table S3—Matrix of genotypic (above the main diagonal) and phenotypic (below the main diagonal) Pearson correlations for the selection of superior individuals.

**Author Contributions:** Conceptualization, M.R.P. and C.E.B.; writing—original draft preparation, M.R.P., J.C.d.R.A., S.R.O.T.d.L., V.T.A., C.E.B., V.C.F. and A.T.B.; writing—review and editing, J.C.d.R.A., V.T.A., C.E.B. and A.T.B. All authors have read and agreed to the published version of the manuscript.

**Funding:** This research was funded by the INCT Café, CNPq, Consorcio Pesquisa Café, and Fapemig.

**Data Availability Statement:** Not applicable.

**Acknowledgments:** The authors would like to thank the Federal University of Lavras, the Research Support Foundation of the State of Minas Gerais (FAPEMIG), the Coordination for the Improvement of Higher Education Personnel (CAPES), and the National Council for Scientific and Technological Development (CNPq) for financial support, as well as the Institute of Natural Sciences, the School of Agricultural Sciences of Lavras, the Graduate Program in Agronomy/Plant Science (Department of Agriculture), the Graduate Program in Genetics and Plant Breeding (Department of Biology), the Minas Gerais Agricultural Research Corporation (Empresa de Pesquisa Agropecuária De Minas Gerais-Epamig), and the National Institute of Coffee Science and Technology—INCT Café.

**Conflicts of Interest:** The authors declare no conflict of interest.

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
