# Peer review of "Multiple Traits Selection Strategies: A Proposal for Coffee Plant Breeding"

_agronomy, doi:10.3390/agronomy13082033_

Round 1
Reviewer 1 Report
The manuscript is relevant to the journal and presented well. I don't have any issues with it. Few minor corrections.
Line 77 – checks?
Line 88 – what country?
Line 101 – seasons?
Line 113- cross out one evaluated
No issues with English
Author Response
Line 77 – checks?
"Chacks" refers to commercial cultivars that were used as controls in the experiment. Changed to "controls".
Line 88 – what country?
In Brazil, corrected in the paper.
Line 101 – seasons?
Yes, corrected in the paper.
Line 113- cross out one evaluated
Corrected in the paper.
Reviewer 2 Report
Review: Multiple traits selection strategies: a proposal for coffee plant breeding
This is a clearly focused, well-written, authoritative ms. I have only one general remark and a few specific comments.
General comment (a suggestion, not a requirement)
I would suggest that the authors provide short explanations for some of the more technical terms and concepts, particularly those not generally used by plant breeders. Some of the statistical methods would benefit from elaboration. The objective would be to make the ms. more accessible, and more easily comprehended by plant breeders and other interested parties.
Specific comments
Page 31
Ms.: “The search for cultivars that are more productive and adapted and resistant to diseases has become a fundamental alternative for more sustainable production and a great challenge for genetic improvement programs.”
Comment: Please consider revising: “adapted and resistant to diseases”. What is meant by “adapted and resistant”; it doesn’t make sense.
Comment: Please consider revising: “has become a fundamental alternative for more sustainable production”. This is neither scientifically correct nor true in any other sense.
Comment: Overall, the objectives for coffee breeding may not include these traits; there are many others that breeders might consider (e.g. pests, drought tolerance, cup profile).
Page 77
Ms.: “Twenty-eight F4:5 progenies and two commercial cultivars as chacks (Catuaí Vermelho IAC 144 and MGS Aranãs) (Supplementary Materials Table S1) obtained by the breeding program at the Agricultural Research Corporation of Minas Gerais (Empresa de Pesquisa Agropecuária de Minas Gerais - Epamig) were used for the evaluation.
Comment: ‘chacks’. Do you mean ‘checks’? Would it be better to say ‘controls’? Please update.
Page 101.
Ms.: “The following traits were evaluated in the crop seasons 2019/2020 and 2020/2021 prior to harvest: stem diameter (SD, cm), the number of plagiotropic branches in the main…”
Comment: Please add whereabouts the stem diameter was measured, e.g. was it at the base?
Page 108
Ms.: “PROD was calculated based on the total fruit harvest, followed by the conversion to bags ha-1 of processed coffee, according to the actual yield of each genotype.”
Comment. Please state the bag size (presumably 60 kg), or preferably state kg/ha.
Only minor editing required.
Author Response
Consulte o anexo

Reviewer 3 Report
In this study, BLUP is used to estimate the variance components. BLUP has been known a good choice for genomic prediction. Do you think if other newly proposed genomic selection methods can achieve better results than BLUP?
In line 141, is the matrix Z the design matrix for markers? Some of the details should be added, such as the distribution of the errors.
In Line 145, p is the environmental effects. Could you explain why the effects were random effects?
Minor editing of English language required.
Reviewer 4 Report
The article proposes a new strategy for coffee breeding. Very innovative. The article demonstrates the feasibility of the strategy through multiple aspects, with detailed data and reliable results.
There is only one small suggestion:to add field photos of coffee so that readers can better understand the data in the article.
Line24-26: The sentence is too long to understand.
Line32-34: two “and”?
Line35: “the need to “ suggest “the need of ”
line39-42:The sentence is too long to understand.
Line364: “as well as “ suggest “As well as ”
